# Transcriptome Profiling of Cardiac Glycoside Treatment Reveals EGR1 and Downstream Proteins of MAPK/ERK Signaling Pathway in Human Breast Cancer Cells

**DOI:** 10.3390/ijms242115922

**Published:** 2023-11-02

**Authors:** Honey Pavithran, Ranjith Kumavath, Preetam Ghosh

**Affiliations:** 1Department of Genomic Science, School of Biological Sciences, Central University of Kerala, Kasaragod 671320, India; honeypavithran6@gmail.com; 2Department of Biotechnology, School of Life Sciences, Pondicherry University, Puducherry 605014, India; 3Department of Computer Science, Virginia Commonwealth University, Richmond, VA 23284, USA; pghosh@vcu.edu

**Keywords:** breast cancer, RNA-Seq analysis, lanatoside C, peruvoside, strophanthidin, anticancer activity, immunoregulation

## Abstract

Cardiac glycosides (CGs) constitute a group of steroid-like compounds renowned for their effectiveness in treating cardiovascular ailments. In recent times, there has been growing recognition of their potential use as drug leads in cancer treatment. In our prior research, we identified three highly promising CG compounds, namely lanatoside C (LC), peruvoside (PS), and strophanthidin (STR), which exhibited significant antitumor effects in lung, liver, and breast cancer cell lines. In this study, we investigated the therapeutic response of these CGs, with a particular focus on the MCF-7 breast cancer cell line. We conducted transcriptomic profiling and further validated the gene and protein expression changes induced by treatment through qRT-PCR, immunoblotting, and immunocytochemical analysis. Additionally, we demonstrated the interactions between the ligands and target proteins using the molecular docking approach. The transcriptome analysis revealed a cluster of genes with potential therapeutic targets involved in cytotoxicity, immunomodulation, and tumor-suppressor pathways. Subsequently, we focused on cross-validating the ten most significantly expressed genes, *EGR1*, *MAPK1*, *p53*, *CCNK*, *CASP9*, *BCL2L1*, *CDK7*, *CDK2*, *CDK2AP1*, and *CDKN1A*, through qRT-PCR, and their by confirming the consistent expression pattern with RNA-Seq data. Notably, among the most variable genes, we identified EGR1, the downstream effector of the MAPK signaling pathway, which performs the regulatory function in cell proliferation, tumor invasion, and immune regulation. Furthermore, we substantiated the influence of CG compounds on translational processes, resulting in an alteration in protein expression upon treatment. An additional analysis of ligand–protein interactions provided further evidence of the robust binding affinity between LC, PS, and STR and their respective protein targets. These findings underscore the intense anticancer activity of the investigated CGs, shedding light on potential target genes and elucidating the probable mechanism of action of CGs in breast cancer.

## 1. Introduction

Breast cancer (BC) continues to be the most common cancer in women worldwide. A comprehensive global cancer survey reported a significant shift, with breast cancer surpassing lung cancer in terms of the highest incidence in 2020 [1]. Thus, there is an urgent need to curb this growing crisis of breast cancer incidence and associated mortality. Over the years, significant advancements have been made in treatment strategies, substantially improving patients’ life expectancy. However, BC is a heterogeneous disease with a high degree of variance in cellular, genome, and transcriptome profiles [2], which affects its progression by impeding the therapeutic response. Therefore, there is a high demand for standard therapeutic options analyzing biological mechanisms and tracing molecular footprints in the disease [3,4]. Several researchers worldwide are focused on developing precision therapies utilizing molecules that specifically target malignant cells. A significant challenge in developing novel compounds lies in the expected time redundancy and resources required to conduct comprehensive research and navigate the complex regulatory processes. This draws attention to the importance of drug repurposing, which uses existing or approved drugs to treat a different ailment, defining newer targets and mechanisms [5].

Cardiac glycosides are an emerging class of compounds that have demonstrated potential anticancer activity by selectively targeting oncogenic proteins and signaling pathways, which makes them valuable candidates for drug repurposing in the treatment of cancer [6,7]. With a history of over a century, these naturally derived compounds have been well-accepted medications for heart failure and arrhythmic disorders. The compounds within this family are notably recognized for their ability to bind and inhibit Na+/K+-ATPase pumps regulating various cellular processes [8]. A decade of numerous intensive studies has been conducted to unravel the potential use of these compounds as front-line cancer medicines. However, the last five years have witnessed a remarkable surge in global interest in unveiling the anticancer activity of cardiac glycosides.

Extensively studied glycosides with promising potential as anticancer agents include digitoxin, digoxin, bufalin, ouabain, peruvoside, and lanatoside C. Ouabain is currently undergoing clinical trials to evaluate its effectiveness in cancer treatment [9]. Assessing the anticancer mechanism triggered by CGs includes several hypotheses and aspects, including modulation of signal transduction, inhibition of angiogenesis, immunoregulation, induction of apoptosis, and cell cycle arrest [10,11]. A central shared characteristic of all CGs is their ability to inhibit the Na/K-ATPase pump, and several theories suggest that these compounds induce cellular cytotoxicity through this primary target [12]. Nevertheless, an understanding of the influential downstream genes and a central regulatory mechanism driving cancer cell death is still lacking.

Our prior research primarily focused on investigating the anticancer efficacy of CGs. In these studies, we have conducted a broad screening of CGs to identify the most promising anticancer compounds through in silico and in vitro studies in three human cancer cell lines, specifically MCF-7 breast cancer cell line, A549 lung cancer cell line, and HepG2 liver cancer cell line. The study screened out three efficient compounds, including lanatoside C (LC), peruvoside (PS), and strophanthidin (STR), with spectacular cytotoxic effects on the mentioned cancer cells with an IC_50_ ranging from 100 nM to 2 µM concentrations. The compounds showed significant anticancer activity, substantiated by comprehensive validations from biochemical assays, molecular analysis, and gene and protein expression assessments. The study concluded by mentioning a considerable influence of MAPK/PI3K/AKT/mTOR signaling pathway regulation upon treatment [13,14,15]. The findings strongly support the hypothesis that LC, PS, and STR can effectively impede cancer proliferation by modulating key proteins involved in vital cellular processes, including cell cycle regulation, apoptosis, and autophagy.

Irrespective of the growing body of evidence regarding the potent anticancer efficacy of CGs, the lack of comprehensive data on their mode of action and genome expression prevents their characterization as effective cancer drugs. In order to address this limitation and delve deeper into the details of our earlier observations, we performed RNA-seq analysis on MCF-7 breast cancer cells treated with LC, PS, and STR. We compared its effects with the untreated control sample. Furthermore, we validated the obtained transcriptome results using qRT-PCR. Following RNA-seq confirmation at the transcript expression level, we attempted to study the interactions between the compounds and crucial proteins using the molecular docking technique. Thus, the current study holds significant importance as it provides transcriptomic data analyzing sequence information and insights into the genes and pathways influenced by CG treatment.

## 2. Results

### 2.1. Transcriptome Profiling of Cardiac Glycoside Induction in MCF-7 Cells Reveals Variability in Genes Associated with Cytotoxicity

To evaluate transcriptional alteration induced by CGs in breast cancer, we performed RNA sequencing on MCF-7 cell lines corresponding to the untreated sample as a control group and the treatment groups, consisting of LC, PS, and STR. The effect of individual treatments compared to control samples is depicted through volcano plots in Figure 1A–C, showing LC vs. Control (CTRL), PS vs. CTRL, and STR vs. CTRL, respectively. LC treatment resulted in 6135 differentially expressed genes, of which 2311 are downregulated and 3824 are upregulated. PS treatment differentially expressed 6079 genes compared to the control, including 2643 downregulated and 3436 upregulated genes. A similar result was observed on treatment with STR, whereby 6143 genes were differentially expressed, of which 2500 were downregulated and 3643 were upregulated. The degree of expressional changes in the control and treatment group is illustrated through a heat map (Figure 1D), where each row represents genes while the column indicates samples. Subsequently, all three treatments were compared to identify the genes that are commonly differentially expressed by LC, PS, and STR groups (Figure 1E). For further analysis, we have compiled and pruned 5813 differentially expressed genes common to all three treatments for downstream functional enrichment analysis.

### 2.2. The Biological Significance of DEGs in P53, MAPK, and Immune Regulatory Pathway

We further performed functional enrichment analysis with the common DEG of the LC, PS, and STR treatment to evaluate the biological relevance of the traced genes through the FunRich algorithm. First, we visualized the gene–gene interactions between the DEG and the enriched genes in the associated pathways, as depicted in Figure 2. This paved the way for understanding the fundamental role of the *TP53* (*P53*) gene, which represented a central regulatory role in integrating multiple interaction networks.

Additionally, we carried out a detailed exploration of the data to gain deeper biological insights into the DEG through functional enrichment analysis of GO terms, including cellular component or localization (CC), biological process (BP), and molecular function (MF). The study revealed that the genes played a significant role in transcription regulation, cell growth, cell cycle progression, immune response modulation, and apoptosis (Figure 3A,B). A detailed analysis of each treatment’s biological influence is shown in Appendix A. We then performed KEGG pathway analysis on the upregulated DE genes, demonstrating their involvement in the P53 pathway, apoptotic pathway, and immune system. Likewise, the corresponding KEGG analysis of the downregulated DE genes revealed their participation in the p38-MAPK pathway, PI3K signaling events mediated by Akt, immune system, and suppression of ER receptors on the plasma membrane (Figure 3C). We then identified a set of ten common Significantly Differentially Expressed (SDE) Genes, namely *EGR1*, *MAPK1*, *CCNK*, *CASP9*, *BCL2L1*, *CDK2*, *CDK7*, *CDK2AP1*, *CDKN1A*, and *P53*, that were involved in regulating these enriched signaling pathways (Table 1). According to whole transcriptome analysis, we found that genes associated with cell cytotoxicity, cell growth, and differentiation were amongst the most highly differentially expressed on treatment with CG. We previously reported that CGs induce cell cycle arrest by attenuating MAPK, PI3K/AKT/mTOR signaling pathways in cancer cells [13,14,15]. Confirming our previous findings, we observed transcriptional changes and influential cytotoxicity-related genes associated with the MAPK/PI3K pathway and P53-dependent apoptosis pathways through RNA-Seq analysis.

### 2.3. Protein–Protein Interaction (PPI) Analysis

The analysis of gene interaction profiles and protein–protein interrelations of SDE genes was carried out using the STRING platform and further explored using Cytoscape software (version 3.9.0). The 10 differentially expressed genes were queried through STRING, and we retrieved molecular networks and functional associations from curated pathways. Additionally, the correlation of DEG was analyzed using the k-means clustering method. The analysis provided two cluster groups, the cyclin-dependent kinase proteins, and the cytotoxic-associated pathways, as depicted in Figure 4.

### 2.4. Confirmation of RNA-Seq Data through qRT-PCR

To further validate the results of RNA-Seq analysis, qRT-PCR assay was performed for the 10 SDE genes, *EGR1*, *MAPK1*, *CCNK*, *CASP9*, *BCL2L1*, *CDK7*, *CDK2*, *CDKN1A*, *CDK2AP1*, and *P53*. The outcomes of the RT-PCR analysis mirrored those of RNA-Seq data, demonstrating that the cardiac glycoside treatment enhanced the expression of *CASP9*, *BCL2L1* (*BAX*), *CDKN1A* (*P21*), and *P53* while turning down the expression of *EGR1*, *MAPK1*, *CCNK*, *CDK7*, *CDK2*, and *CDK2AP1* genes, as shown in Figure 5.

### 2.5. Cardiac Glycosides Exhibit Cell Growth Arrest in Breast Cancer Cells Altering p53 Dependent MAPK Signaling Pathway

In cancer cells, MAPK signaling facilitates mitosis entry and extends cell growth and division while ensuring the downregulation of cancer suppressor and pro-apoptotic proteins such as p53 and caspase-9 [16]. To elucidate the interdependence of MAPK, p53, and apoptotic signaling pathways in response to cardiac glycoside treatment, we performed Western blot analysis to quantify protein expression. The findings revealed significant downregulation of MAPK1 protein, and its downstream effector protein EGR1, upon treatment (Figure 6). This reduction likely influences the activation of the cyclin cascade of proteins, subsequently leading to cell death. Additionally, we observed upregulation of the p53 and caspase-9 protein safeguarding cells to undergo cell death. Over the years, it has become evident that the transcription factor EGR1 directly controls several tumor suppressor genes like PTEN and p53 [17]. The protein expression analysis confirms the specificity of the tested compounds in inhibiting MAPK and associated signaling cascades, thereby turning off EGR1 gene expression.

### 2.6. Cardiac Glycosides Effectively Downregulated EGR1 Expression Via Influencing MAPK Signaling Cascade Controlling Cell Fate Decision

To provide further evidence supporting the dependence of the MAPK pathway and the p53 pathway in regulating cell growth, we performed immunostaining of MAPK1, p53, and EGR1 proteins studying the co-regulation and expressional changes on treatment with CG. We prioritized the key proteins of MAPK signaling, the p53 pathway, and downstream effectors through a combination of our transcriptomic analysis and subsequent assays. The fluorescent image (Figure 7A) shows that the level of MAPK1 protein is significantly reduced on three treatments compared to control cells. Whereas, p53 has lower expression in control than treatments (Figure 7B). Figure 7C clearly depicts the effect of CG on MCF-7 cells, where the cytoplasmic expression of EGR1 protein is highly reduced upon treatment. The analysis revealed that cardiac glycoside treatment significantly reduced the expression of MAPK1 and the downstream effector protein EGR1. The downregulation of MAPK1 protein is further promoted by p53 expression, suggesting that MAPK dynamics is crucial to establishing a balance between cell cycle arrest and cell progression in a p53-dependent manner.

### 2.7. Protein–Ligand Interaction Analysis

Following the identification and confirmation of crucial genes influencing treatment outcomes with LC, PS, and STR, there was a compelling interest in further elucidating the precise impact of these compounds on proteins. Molecular docking studies were conducted to investigate this phenomenon to understand the interaction between the compounds and the protein units of the 10 scrutinized genes. The docking studies revealed higher binding efficiency and dominant bond interactions between the compounds and proteins (Table 2). Our particular focus was on comprehending the MAPK-dependent activation of EGR1 in driving transcriptional regulation and subsequently controlling cell proliferation and growth in cancer cells. This significant focus on evaluating the interactions between the EGR1 protein and the ligands revealed a robust binding efficiency, with binding scores of −8.8, −6.8, and −7.2 kcal/mol for LC, PS, and STR, respectively. The active site binding and the major residues involved in the interaction between ligands and EGR1 protein are depicted in Figure 8A–F. Similarly, MAPK1 protein showed significant ligand interaction with binding scores of −9.6, −8.2, and −7.3 kcal/mol for LC, PS, and STR, respectively (Figure 8K–L). The higher binding efficiency of LC with the proteins compared to the other ligands is probably due to its ability to engage in a broader range of H-bond interactions. The docking images and the corresponding interacting residues of p53, CCNK, CASP9, BCL2L1, CDK7, CDK2, CDK2AP1, and CDKN1A protein–ligand interactions are shown in Appendix A.

## 3. Discussion

Drug repurposing is an accelerated drug development approach that involves the identification of novel clinical possibilities of existing drugs, thereby bypassing prolonged traditional drug development methods. The concept has been effectively studied and utilized across a spectrum of medical conditions combating ailments like viral infections [18], contagious COVID-19 [19], influenza infection [20], antimicrobial resistance [21], etc. The therapeutic strategy was practiced and implemented in cancer treatment for many years. Over time, several candidate compounds have been successfully repositioned, including Doxorubicin, Cyclophosphamide, Everolimus, Tamoxifen, and Anastrozole, as remedial drugs to treat various cancers [22].

Cardiac glycosides are among the promising candidates studied and tested through drug repurposing approaches and have demonstrated widespread utility for anticancer and antiviral activities [23]. Notably, during the last 15 years, compelling evidence has increasingly supported the potency of CGs as a potent antiproliferative agent in various forms of cancer [24,25]. Earlier studies have demonstrated the antitumor efficacy, immunomodulatory effects, ability to induce apoptosis, and other notable activities of more than 20 such compounds [26]. Recent research focusing on the anticancer effect of ouabain has uncovered its senolytic property, adding another piece to the spectrum of potential applications of CGs [6,27]. Inspired by this work, our research team has been committed to exploring the efficacy of CGs in combating cancer. Through rigorous in silico and in vitro screening processes, we have identified three potent members of the family, namely Lanatoside C, Peruvoside, and Strophanthidin, with high efficacy in various human cancers, including lung, liver, and breast cancer. Further analysis of each molecule has revealed their antiproliferative mechanism, which involved attenuation of the MAPK, PI3K/AKT/mTOR, and Wnt/β-catenin signaling pathways, causing apoptosis induction and cell cycle arrest [13,14,15,28,29,30].

In the current study, we performed comprehensive RNA sequencing to gain more information on the transcriptional regulation and mechanism of action responsible for the LC, PS, and STR anticancer activity in the MCF-7 cell line. For this purpose, we sequenced four samples of MCF-7 cells, including three treatments (LC, PS, and STR) and an untreated control sample. The RNA-Seq analysis of the corresponding sequence data identified 5813 DEGs common in the three treatment groups compared to the control sample. We also identified a significant correlation between the MAPK1 and EGR1 genes, which are associated with cell growth during differential gene expression analysis, following treatment with CGs (Figure 1D). Additionally, functional enrichment analysis of the curated common genes showed their involvement in oncogenesis, cell cycle regulation, apoptosis, and complex interconnections of associated signaling pathways in immunoregulation. Accumulating evidence supports the efficacy of CGs as potent anti-inflammatory agents in various ailments, including neurodegenerative diseases [31], autoimmune diseases [32], and cancer [11,33] via influencing cellular signaling pathways. Recently, Škubník et al. discussed the broad immunomodulatory actions of CGs in cancer treatment, influencing immune system regulators and thereby inhibiting the transcriptional activity of key oncogenes [34]. Using the FunRich algorithm, we were able to infer the central role of p53 gene alteration during treatment (Figure 2). Moreover, the functional enrichment analysis of upregulated genes revealed them to be members of the P53-dependent apoptosis pathway. Yet another essential finding derived from the transcriptome analysis was the differential expression of cell cycle regulators, including CDK2 (Cyclin Dependent Kinase-2), CDK2AP1 (Cyclin Dependent Kinase-2 Associated Protein), CDK7 (Cyclin Dependent Kinase-7), and CCNK (Cyclin K), pointing towards the potential link between cell growth and the apoptosis pathway through immune regulation. However, the role of *CDK2AP1* is still ambiguous, as it has a complex role in different types of cancer. While its exact function is still not fully understood, there are indications of its tumor suppressive activity in breast cancer cells via regulating cell cycle proteins [28]. A recent study on hepatocellular carcinoma (HCC) marked the gene as a prognostic indicator, as its expression was significantly lower in normal tissue than in tumor tissues. These findings confirmed the metastatic effect of CDK2AP1protein and highlighted the potential of CDK2AP1 as a target for immunotherapy in HCC [35].

The transcriptome profiling further validated our previous findings, demonstrating significant downregulation of key genes associated with MAPK- PI3K oncogenic signaling pathways on treatment with CGs. However, the downstream effectors responsible for the cytolysis of cancer cells on treatment with CGs, have not been previously identified. Following the identification of crucial genes associated with cancer, we quantitated the variability in mRNA expression through qRT-PCR (Figure 5). We then extended our investigations to assess the impact of the selected CGs on translational processes using immunoblotting (Figure 6) and immunocytochemical analysis (Figure 7). Our findings revealed significant alterations in the expression of the target proteins upon treatment with CGs. Notably, we observed a coordinated downregulation of the MAPK pathway and an upregulation of p53 expression, which jointly exerted control over cell progression at the translational level. These results shed light on the dynamic interplay between MAPK and p53 within cancer cells and highlight the potential for targeted interventions using CGs. In particular, our observations pointed to substantial suppression of EGR1, a downstream transcription factor activated by the MAPK-ERK signaling pathway in cancer. It is widely accepted that induction of the p38-MAPK pathway leads to activation of EGR1 through cyclic AMP-response element binding protein (CREB) transcription factor [29]. A recent report by Wang et al. highlighted the significance of the EGR1 transcription factor in cancer. The authors illustrated the impact of the EGR1 gene in tumor initiation, cell death, immune response, and its potential role in the tumor microenvironment. The significance of the EGR1 protein as the downstream regulator of the MAPK signaling pathway gains higher importance as it can be targeted for cancer therapy [30,36]. Studies have also indicated that EGR1 plays a regulatory role in angiogenic and osteoclastogenic factors, which contribute to metastasis by influencing tumor suppressors such as p53 and PTEN [37,38]. A recent study by Shan reported the significant effect of *EGR1* response interconnecting three principal signaling pathways. This study provides essential evidence for an interdependent effect of endoplasmic reticulum stress on the MAPK pathway, transcription regulation through the *EGR1* gene, and cell death mechanism in HepG2 cells [39]. At the same time, a similar study reported *EGR1* expression through the *p38* MAPK pathway modulating immune response and finally limiting transcription [40]. Previous works identified EGR1 as a target gene getting promptly activated by various mitogens, and apoptotic signaling pathways [41] in various cancers including lung [42], prostate [43], and breast [44]. Thus, our results discuss a concordance similar to these studies, where we report the influential role of cell cycle key regulators, *CDKs*, MAPK pathway genes, and apoptosis regulators on treatment with cardiac glycosides.

Through the utilization of RNA-Seq data, we have substantially enhanced our understanding of the potential mechanistic action of CGs on breast cancer cell lines. We have identified differentially expressed genes with substantial relevance to breast cancer, thereby augmenting our understanding of their significant roles. In addition to exploring the transcriptome, we have uncovered the translational effect and critical pathway interconnections that underlie the cytolytic properties of the compounds. These results also suggest that EGR1 dysregulation defines the inflammatory and immunosuppressive effects in breast cancer, causing loss of cell cycle control and apoptosis induction [45,46,47]. This study identifies the downstream effector genes and transcriptional regulators that contribute cytotoxic properties to the compound. These findings have enabled us to pinpoint potential targets that may exert a significant impact on CG treatment in breast cancer.

## 4. Materials and Methods

### 4.1. Cell Culture Reagents and Chemicals

A human breast cancer cell line (MCF-7) was obtained from the National Centre for Cell Science (NCCS), Pune, India. The obtained MCF-7 cells were cultured in DMEM medium (Himedia) supplemented with 10% fetal bovine serum (Gibco; Thermo Fisher Scientific, Inc., Cambridge, MA, USA), 100 µ/mL antibiotic–antimycotic solution (Himedia Pvt., Ltd., Mumbai, India) and maintained at 37 °C with 5% CO_2_. The CG compounds, including lanatoside C and strophanthidin, were purchased from Sigma Aldrich (St. Louis, MO, USA), while peruvoside was procured from Toronto Research Chemicals (North York, ON, Canada) and dissolved in dimethyl sulphoxide (DMSO) by maintaining the overall DMSO concentration not exceeding 0.01%.

### 4.2. RNA Isolation and Quality Assessment

The MCF-7 cells were seeded in 6-well plates to isolate RNA and divided into control and treatment groups. The treatment group was exposed to lethal concentrations of IC_50_ of CGs, determined based on our previously published studies, specifically, LC (1.2 µM), PS (100 nM), and STR (2 µM), respectively, and incubated for 24 h at 37 °C. After incubation, total RNA was isolated using TRIzol^TM^ reagent (Invitrogen, Waltham, MA, USA.; Thermo Fisher Scientific, Inc., Boston, MA, USA) following the manufacturer’s instructions. Next, the concentration and purity of isolates were measured using a NanoDrop 2000C spectrophotometer (Thermo Scientific, Boston, MA, USA), and the isolated total RNA was then subjected to sequencing to obtain expression data.

### 4.3. Library Preparation and Sequencing

The genome-wide transcriptome library preparation and high-throughput RNA sequencing were performed at Eurofins Scientific (Bangalore, India). The extracted total RNA was used for preparing sequence-read libraries using the Illumina TrueSeq RNA sample prep kit. Subsequently, quality assessment and quantification of the prepared RNA libraries were evaluated using an Agilent 2100 bioanalyzer (Agilent, Santa Clara, CA, USA). The samples were also sequenced on the Illumina HiSeq250 platform, obtaining 100 bp paired-end (PE) sequences with a mean sequencing depth of 45 million reads per sample.

### 4.4. Transcriptome Data Analysis

The quality control and preprocessing of raw sequence data acquired from high-throughput sequencing pipelines were performed using FastQC and fastp [48] algorithms, which formulated the data for downstream analysis. After filtering out low-quality reads, raw reads were mapped to the human reference transcriptome (Ensembl; Homo sapiens version 86) using Kallisto version v0.48.0. [24]. The subsequent analyses were performed using the statistical computing environment, R version 4.2.1, R studio version 2022.07.2, and Bioconductor, version 3.14 [49]. Transcript quantification data were summarized to genes using TxImport packages [50] and normalized using the TMM method in EdgeR [51]. The normalized, filtered count data were variance stabilized using the voom (variance modeling at the observational level) function in EdgeR. Subsequently, the most exact process in EdgeR was used to perform differential gene expression analysis based on a negative binomial generalized linear model (GLM) applied to the count data (FDR ≤ 0.01, logFC ≥ 1).

### 4.5. Gene Set Enrichment and Gene Interaction Network Analysis

The commonly regulated differentially expressed genes on treatment with LC, PS, and STR were subjected to functional annotation and gene interaction network analysis using the FunRich algorithm [52]. FunRich performs comprehensive set functional enrichment, gaining biological insights from high-throughput experiments to identify overrepresented classes of genes.

### 4.6. Protein–Protein Interaction (PPI) Analysis

Protein–protein interactions are some of the crucial components that determine the cellular processes involved in normal as well as disease conditions. We used the STRING algorithm [53] to analyze protein interactions and assess the functional role of significant proteins from curated pathways. The STRING database provides known and predicted reliable protein–protein associations based on co-expression analysis, evolutionary signals, and ortholog-based evidence across organisms.

### 4.7. Quantitative Real-Time PCR Analysis

Quantitative real-time PCR was used to validate the selected differentially expressed (DE) genes retrieved from RNASeq analysis on treatment with cardiac glycosides. cDNA was synthesized from 2 µg total RNA using a verso cDNA synthesis kit (Thermo Fisher Scientific, Inc.) following the manufacturer’s instructions. The reaction mixture included 10 µL of 2X SYBR green qPCR master mix (Kapa Biosystems, Wilmington, MA, USA), nuclease-free water (Himedia), 100 ng RNA samples, and 100 ng RT primers. The primers were designed using Primer 3 and are listed in Appendix A. The PCR reactions were performed on the Roche light cycler^®^ 480 system (Roche diagnostic corporation, Indianapolis, IN, USA) following specific experimental conditions, an initial denaturation step at 95 °C for 10 min, followed by 35 cycles of denaturation at 95 °C for 30 s, annealing at 55 °C for 15 s, and extension at 72 °C for 15 s. This was followed by a final extension step at 95 °C for 1 min and a final annealing step at 60 °C for 1 min. The relative expression of each gene was normalized with an internal control β-actin and was calculated using the 2^−ΔΔCT^ method. The experiments were further replicated in triplicate to ensure the accuracy and reliability of the results.

### 4.8. Immunoblotting Analysis

The effect of specific CGs at the translation level in a breast cancer model, MCF7 cells was assessed through immunoblotting or Western blot analysis. After a 24 h incubation period with the treatment, the cells were lysed in RIPA buffer containing a protease inhibitor cocktail (Merck, Darmstadt, Germany). The total protein concentrations were determined using the Bradford assay, and 20 µg of protein was loaded into each well of a 10% SDS-PAGE gel. After the electrophoresis, the gel was transferred to a PVDF membrane (Merck Millipore) using the Trans-Blot^®^ Turbo TM blotting system (Bio-Rad, Hercules, CA, USA). The membrane was then blocked with a 3% Bovine Serum Albumin (BSA) solution in 1X Tris Buffered Saline-Tween (TBST) and subsequently incubated with the primary antibody overnight at 4 °C. Subsequently, the membrane was washed 3 to 4 times with 1X TBST and then incubated with the appropriate secondary antibody for an hour at room temperature. The primary antibodies used in this study were EGR1 Rabbit polyclonal antibody, MAPK1/ERK2 rabbit polyclonal antibody, Caspas-9 mouse polyclonal antibody, and p53 Rabbit polyclonal antibody (Origin India Pvt., Ltd., Mumbai, India). The chemiluminescent signal was detected and analyzed using a C-Digit chemiluminescent western blot scanner (LI-COR, Lincoln, NE, USA).

### 4.9. Immunocytochemical Analysis of Protein Co-Localization

For the Assay, the MCF-7 cells were plated on a coverslip in 6-well plates with a density of 2 × 104 cells per well. The cells were incubated at 37 °C with 5% CO_2_ and further treated with lethal doses of lanatoside C, peruvoside, and strophanthin for 24 h. After the incubation period, the media containing the compounds was aspirated and the cells were washed with 1 × PBS (Phosphate Buffer Saline) about 3 times. The cells were then fixed with 4% paraformaldehyde, followed by permeabilization with 0.1% of Triton x-100 for 5 min, and further blocked with 3% BSA for an hour at room temperature. Afterward, the cells were incubated with the following primary antibodies, EGR1 Rabbit polyclonal antibody, MAPK1/ERK2 rabbit polyclonal antibody, and p53 Rabbit polyclonal antibody (Origin, India) overnight at 4 °C. The cells were further washed with 1× PBS and incubated in secondary antibody anti-mouse AlexaFluor-555/anti-rabbit AlexaFluor-488 for an hour at room temperature. To determine protein colocalization within cells, 0.5 µg mL^−1^ of DAPI was used to incubate cells in the dark for 10 min. The coverslips were subsequently mounted on glass slides coated with ProLong Gold Antifade Mountant and examined using a fluorescent microscope at a Magnification of 100× while also capturing images (Zeiss, Axio Observer, Oberkochen, Germany).

### 4.10. Molecular Docking

Molecular docking studies were performed to evaluate and validate the interaction of CG compounds with scrutinized gene products. To this end, the ligand and candidate target protein structures were extracted from PubChem (https://pubchem.ncbi.nlm.nih.gov/, accessed on 25 June 2023) and RSCB protein data bank (https://www.rcsb.org/, accessed on 25 June 2023), respectively. The molecular docking was followed in PyRx version 0.9 [54] using the Auto-dock vina program. Subsequently, the docking results were analyzed and visualized using PyMol 2.5 and Discovery Studio Visualizer (v21.1.0.20298). The best docking confirmations were selected from 10 predicted poses, comparing the ranked binding affinity score and RMSD values.

### 4.11. Statistical Analysis

Differential gene expression (DEG) analysis and its accountability were tested in EdgeR. The systematic evaluation and interpretation of significant DEGs were performed in the R statistical environment considering *p* < 0.05. The means of fold change between multiple replicates of RT-qPCR results were compared using GraphPad Prism, version 9, using the analysis of variance (ANOVA) method. Western blot data were compared as mean ± SEM from two independent experiments (*n* = 2). Statistical significance of difference in drug-treated versus control cells was determined using the Student *t*-test.

## 5. Conclusions

In this study, we have substantially expanded our understanding of the effect of CGs in breast cancer by utilizing RNA sequencing. It is also worth noting that the study marks the first instance in which the action of CGs in cancer cell lines is studied through transcriptome-wide analysis of differentially expressed genes. Through the study, we obtained confirmatory results that supported previous findings, which showed the significance of the repression of cell proliferation through the MAPK/PI3K/AKT pathway and its influence on cell cycle and apoptosis. In addition, we found differential expression of downstream genes and transcription factors associated with the MAPK-PI3K pathway. The analysis provided evidence for the regulation of EGR1 through MAPK and the P53-dependent signaling cascade, which consecutively mediated cell toxicity (Figure 9). Notably, our findings demonstrate the significant regulation of MAPK, p53-dependent apoptosis, and downstream effectors on treatment with CGs. The importance of these observations is in pointing out the effectiveness of selected CGs in the treatment of breast cancer. In summary, we emphasize the anticancer potency of cardiac glycosides, specifically lanatoside C, peruvoside, and strophanthidin, in controlling cell proliferation, thereby indicating their effectiveness in treating breast cancer.

## Figures and Tables

**Figure 1 ijms-24-15922-f001:**
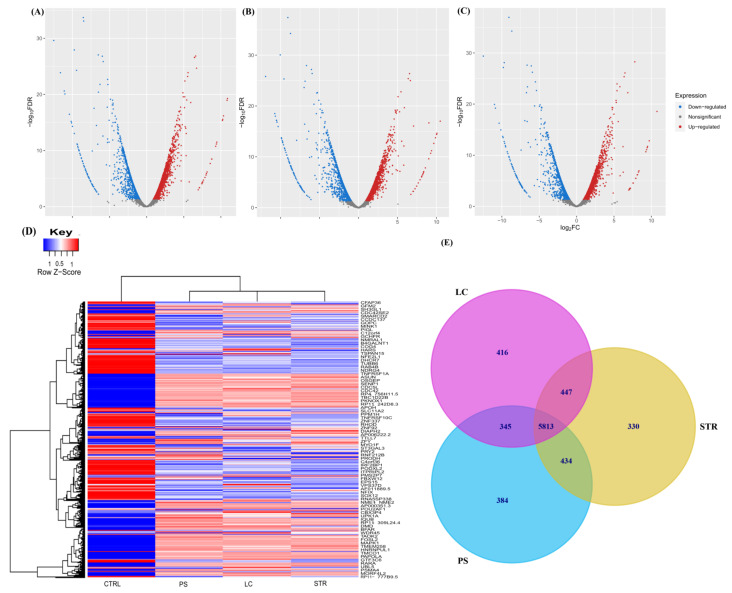
Analysis of differential gene expression from Cardiac-glycoside-treated MCF7 cells using RNA-Seq pipelines. Volcano plot showing differentially expressed genes on treatment relative to control sample, (**A**) Downregulated genes (blue; *n* = 2311), upregulated genes (red; *n* = 3824) on LC treatment. (**B**) Downregulated genes (blue; *n* = 2643) and upregulated genes (red; *n* = 3436) on PS treatment. (**C**) Downregulated genes (blue; *n* = 2500) and upregulated genes (red; *n* = 3643) on STR treatment. (**D**) Heat map representation of normalized and filtered differentially expressed genes between treated (LC, PS, and STR) and untreated MCF7 cells. (**E**) Venn diagram illustrating DEG in LC, PS, and STR.

**Figure 2 ijms-24-15922-f002:**
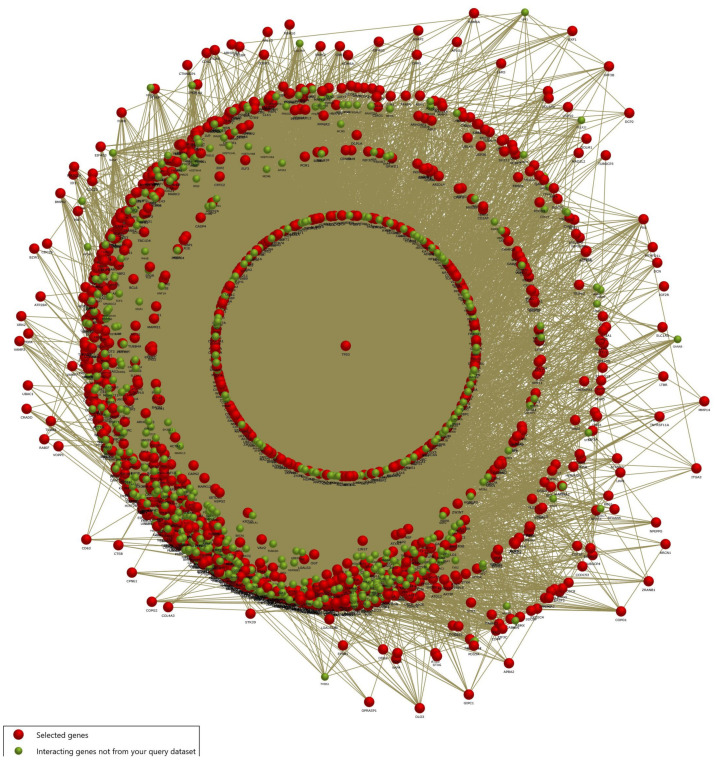
Interactome network of differentially expressed genes common across all treatments (LC, PS, STR). The red node indicates DEGs, while the green node emphasizes interacting genes in enriched pathways from the query.

**Figure 3 ijms-24-15922-f003:**
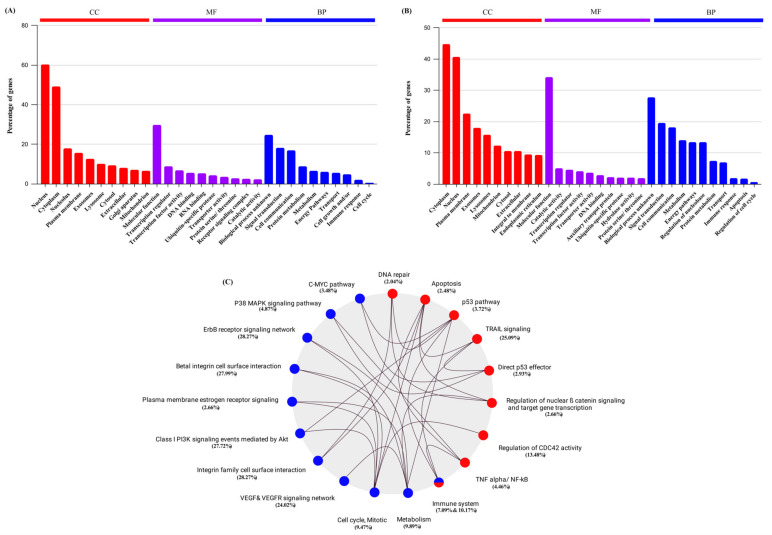
Functional enrichment analysis of Differentially expressed genes. (**A**) Gene ontology (GO) analysis of commonly downregulated genes in LC, PS, and STR treatment. (**B**) Gene ontology (GO) analysis of commonly upregulated genes in LC, PS, and STR treatment. (**C**) KEGG pathway enrichment analysis of commonly downregulated (blue) and upregulated (red) genes in LC, PS, and STR treatment and their interactions.

**Figure 4 ijms-24-15922-f004:**
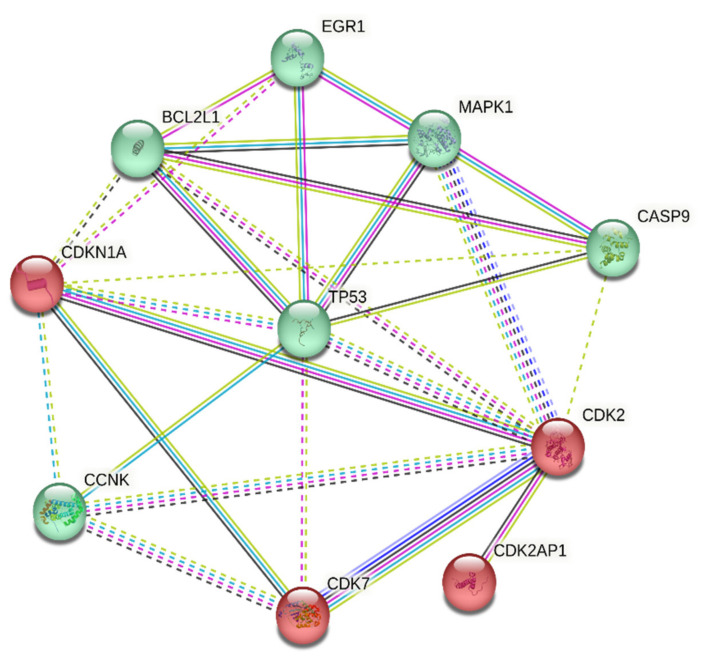
STRING analysis of candidate differentially expressed genes, characterized into two clusters of pathway regulatory mechanism.

**Figure 5 ijms-24-15922-f005:**
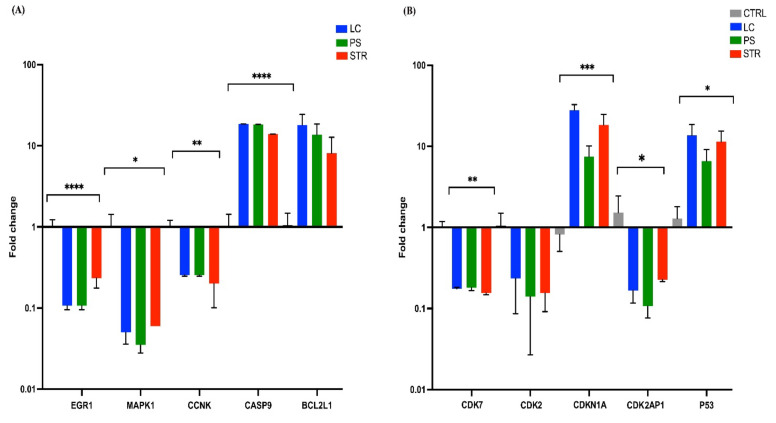
RT-PCR analysis of the 10 candidate gene expressions in control, LC, PS, and STR. (**A**) Bar graph representing the fold change of EGR1, MAPK1, CCNK, CASP9, and BCL2L1 (**B**) Bar graph representing the fold change of CDK7, CDK2, CDKN1A, CDKN2AP1, and P53. All *p*-values are summarized with asterisks (ns *p* > 0.05, * *p* ≤ 0.05, ** *p* ≤ 0.01, *** *p* ≤ 0.001, **** *p* ≤ 0.0001).

**Figure 6 ijms-24-15922-f006:**
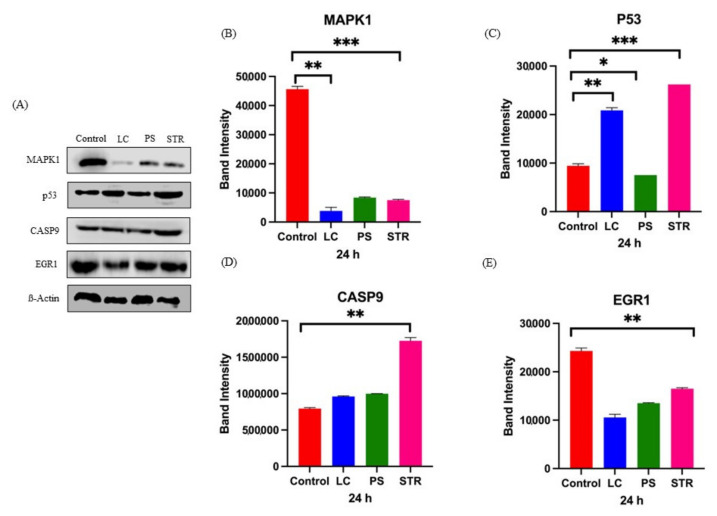
Western blot expression of influenced protein on treatment with cardiac glycosides. (**A**) Representative Western blot showing the expression of crucial influenced proteins including MAPK1, p53, CASP9, and EGR1 on treatment with lanatoside C (LC), peruvoside (PS), and strophanthidin (STR). (**B**) Quantification of Western blot, showing decreased expression of MAPK1 protein on treatment with respective cardiac glycosides. (**C**) Quantification of Western blot, showing increased expression of p53 protein on treatment with respective cardiac glycosides. (**D**) Quantification of Western blot, showing increased expression of caspase-9 protein on treatment with respective cardiac glycosides. (**E**) Quantification of Western blot, showing decreased expression of EGR1 protein on treatment with respective cardiac glycosides. All *p*-values are summarized with asterisks (ns *p* > 0.05, * *p* ≤ 0.05, ** *p* ≤ 0.01, *** *p* ≤ 0.001).

**Figure 7 ijms-24-15922-f007:**
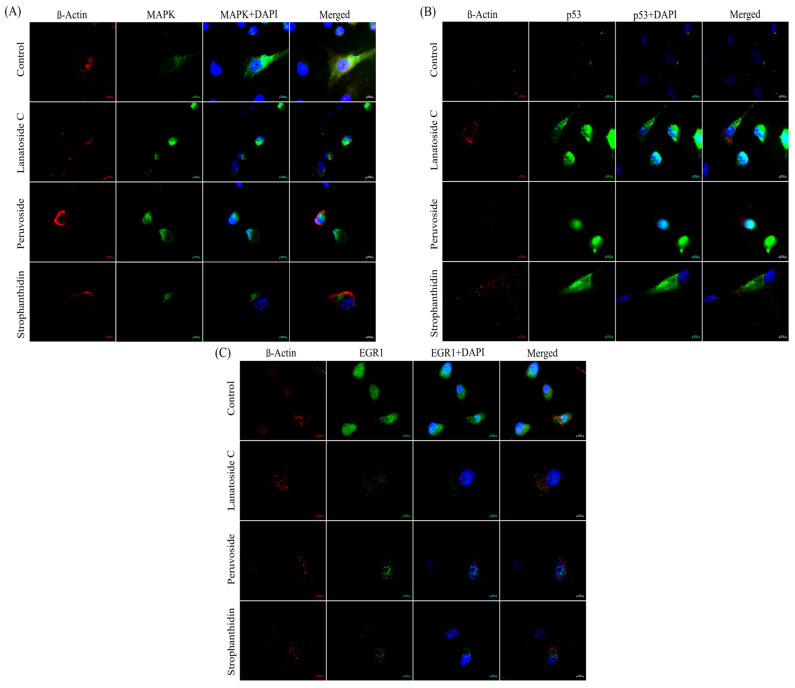
Immunofluorescence imaging of subcellular localization of target proteins in MCF-7 cell lines. (**A**) Inhibition of MAPK1 protein expression (MAPK1 antibody—green) on treatment condition compared to control, where the protein is highly expressed. (**B**) Enhanced expression of p53 protein (p53 antibody- green) on treatment. (**C**) Immunoblotting image analysis of EGR1 protein (EGR1 antibody- green) showing significant downregulation on treatment in accordance with the expressional alteration of MAPK1 protein. The β-Actin antibody is indicated in red colour while the blue colous is hoechst 33342 staining nu-clear portion of the cell.

**Figure 8 ijms-24-15922-f008:**
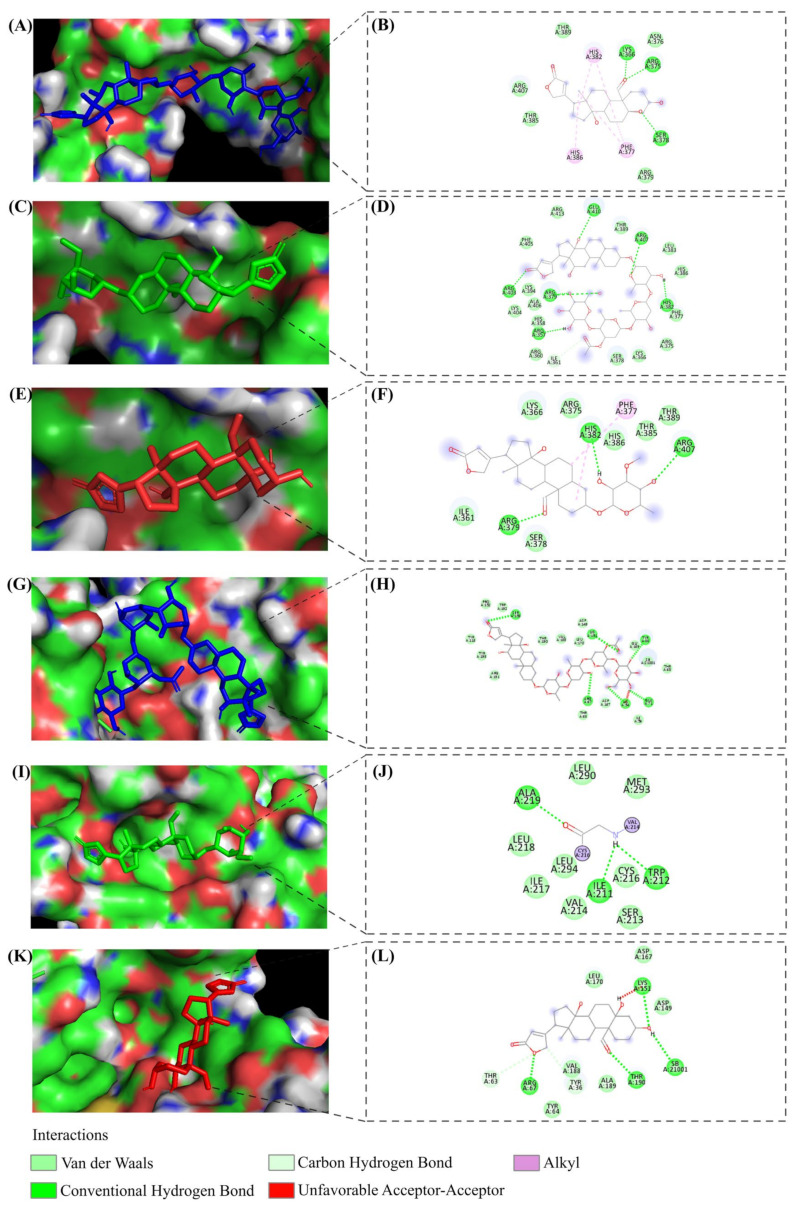
Molecular docking interactions of LC, PS, and STR ligands. (**A**–**F**) Interactions of ligand molecules with EGR1 protein in 3D and 2D images. (**G**–**L**) Interactions of ligand molecules with MAPK1 protein in 3D and 2D images.

**Figure 9 ijms-24-15922-f009:**
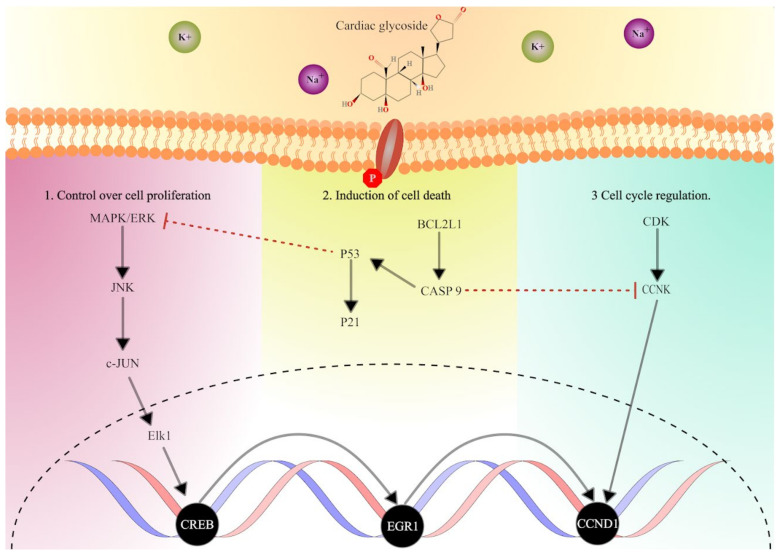
Hypothetical overview of the mechanistic impact of cardiac glycosides on multiple core oncogenic signaling pathways in breast cancer. Speculative examinations suggest that CGs influenced the MAPK pathway, p53 pathway, and CDK pathway, controlling cell growth, apoptosis, cell fate, cell cycle regulation, and cell division, respectively. The schematic representation includes (1) the upstream activators of MAPK/ERK signaling (2) regulation of p53-dependent cell death, and (3) modulators of cell cycle progression and cell division, alongside the downstream effectors of signaling pathways controlling cancer progression.

**Table 1 ijms-24-15922-t001:** Significantly differentially expressed genes selected for downstream analysis and validation.

Gene ID	logFC	Log CPM	*p*-Value	FTR
EGR1	−8.54101	5.887722	3.054027 × 10^−38^	2.010161 × 10^−34^
CDK2	−2.190541	6.222022	4.543721 × 10^−7^	2.986198 × 10^−6^
CDK7	−2.971352	7.432732	2.432360 × 10^−11^	4.535352 × 10^−10^
CDKN2AIP	−2.589947	5.061544	8.223606 × 10^−9^	8.372433 × 10^−8^
CCNK	−2.220707	6.667618	2.973957 × 10^−7^	2.04648 × 10^−6^
MAPK1	−1.575896	6.162307	2.133276 × 10^−4^	7.070103 × 10^−4^
CDKN1A	1.625336	6.646884	0.0001267927	0.0004421453
BCL2L1	1.751856	5.82682000	4.467876 × 10^−5^	0.0001757249
CASP9	1.735179	2.647814	3.450492 × 10^−4^	1.080197 × 10^−3^
TP53	2.372977	5.134613	8.836985 × 10^−8^	7.003616 × 10^−7^

**Table 2 ijms-24-15922-t002:** The docking analysis results, portraying the binding energy score and amino acids interacting between the bonds.

Interacting Protein	Compound	Gibbs Free Energy (Kcal/mol)	Hydrogen Bonding	Van Der Waals Interactions
EGR1	LC	−8.8	LYS 366, ARG 375, SER 378	THR 385, ARG 407, THR 389, ARG379, ASN 376
	PS	−6.8	GLU 410, ARG 407, HIS 382, ARG 379, ARG 357	ILE 361, ARG 360, HIS 358, LYS 404, ALA 406, LYS 394, PHE 405, ARG 413, THR 389, LEU 383, HIS 386, PHE 377, ARG 375, LYS 366, SER 378
	STR	−7.2	HIS 382, ARG 407, ARG 379	LYS 366, ARG 375, HIS 386, THR 385, THR 389, SER 378, ILE 361
MAPK1	LC	−9.6	SER 158, LYS 151, TYR 96, GLU 71, LYS 54, ARG 67	PRO 152, TRP 192, ASP 149, THR 190, VAL 188, LEU 170, GLY 169, THR 68, ILE 56, ASP 167, THR 63ARG 191, TYR 193, TYR 113
	PS	−8.2	ALA 219, ILE 211, TRP 212	LEU 218, LEU 290, ILE 217, LEU 294, VAL 214, SER 213, CYS 216, MET 293
	STR	−7.3	ARG 67, LYS 151, THR 190	THR 63, TYR 64, VAL 188, TYR 36, ALA 189, LEU 170, ASP 167, ASP 149
CCNK	LC	−9.4	VAL 106, LYS 754, GLN 156	LYS 172, GLN 179, LYS 175, LEU 171, ILE 178, LYS 985, VAL 182, ALA 168, PRO 939, LYS 166, ARG 868, GLN 162, GLU 107, ASP 69, LEU 861, GLU 158, HIS 159, VAL 157, PHE 833, GLN 758, GLU 108
	PS	−9.2	ASP 855, GLN 1015	GLU 752, PHE 791, ILE 765, ALA 732, LEU 844, HIS 796, MET 794, TRP 1014, LYS 721, TYR 793, ASP 795, ILE 711, ASP 797, VAL 719, LYS 734, ALA 854
	STR	−9.5	GLN 1015	ASP 795, ASP 797, HIS 796, TYR 793, MET 794, ILE 711, ALA 732, VAL 719, PHE 791, ILE 765, ALA 854, LYS 734, ASP 855, LEU 844
CASP9	LC	−9.5	GLU 261, GLN 245, THR 337, SER 339, PHE 267, GLY 269	LEU 256, LYS 328, SER 242, HIS 302, CYS 303, ARG 258, ASN 265, PRO 338, ASN 268, ILE 341, LYS 280, GLY 276, GLY 277, LEU 335, PHE 246
	PS	−7.6	ILE 396, LYS 398, TYR 397, PHE 351	GLN 399, PHE 348, PRO 357, ARG 355, TRP 354, GLY 350, PRO 349, GLY 395
	STR	−6.9	ASN 265	LYS 299, GLY 305, PHE 319, TYR 324, GLN 320, GLN 245, THR 337, VAL 264, PHE 246, ARG 258, GLY 304, GLY 306
BCL2L1	LC	−8.2	LYS 366, ARG 375, SER 378	HIS 382, HIS 386, PHE 377
	PS	−7.7	GLU 410, ARG 407, HIS 382, ARG 379, ARG 357	
	STR	−6.5	HIS 382, ARG 407, ARG 379	PHE 377
CDK7	LC	−10.8	GLU 95, ASP 97, ASP 137, LYS 139, LYS 41, LYS 103, SER 106,	ASN 311, PRO 310, THR 96, MET 94, VAL 26, ASN 141, GLY 157, HIS 135, SER 161, GLN 22, GLY 21, ASP 155, LEU 18, ASN 105, ASP 104, LEU 107
	PS	−8.2	ASN 142, LYS 41	GLY 19, THR 96, VAL 100, ASP 97, LYS 139, ASP 155, GLY 157, ASP 137, GLN 22, SER 161, PHE 23, LYS 160, GLY 21, LEU 144, VAL 26, LEU 18, GLU 95
	STR	−9.3	MET 94	GLN 172, VAL 26, LYS 41, LEU 18, PHE 91, ASP 92, ILE 75, PHE 93, GLU 95, THR 96, ASP 97, ASN 142, GLY 21, ASN 141, GLU 20, TPO 170 GLU 99
CDK2N1A	LC	−9.2	ARG 103, ALA 134, GLY 139, ARG 131, GLU 149, ARG 144, ARG 107	GLY 135, VAL 115, ARG 138, THR 137, SER 140, ASP 116, ALA 148, ALA 147, GLY 111, ARG 112, ALA 143, ASP 105, VAL 106
	PS	−7.6	TYR 44, GLN 50, MET 54, ARG 87, ASP 84, ARG 46	THR 18, VAL 51, MET 52, MET 53, GLU 88, ALA 21, ARG 22
	STR	−7.2	ARG 87, GLU 88, MET 54, ARG 46	TYR 44, GLY 55, MET 53, MET 52, VAL 51, ASP 84, GLN 50
CDKN1A	LC	−9	ARG 210, ASN 213, ARG 146, GLU 143, LYS 110	ASP 156, HIS 158, SER 152, CYS 148, THR 216, ALA 145, GLU 85, GLY 83, ASN 84, ASP 86, PRO 106, ALA 105, ARG 149, GLY 142, THR 219, LYS 217, LEU 209
	PS	−7.9	ILE 30, GLN 125, LYS 248	TYR 250, HIS 246, GLN 49, THR 51, SER 31, ASN 65, ASP 29, ASN 36, GLN 38, ILE 128, GLU 130
	STR	−7.6	MET 40, SER 39	SER 42, ASP 41, LEU 121, GLU 25, CYS 27, ALA 26, ILE 23, GLU 124, HIS 44, ASP 122, ARG 155
CDK2	LC	−8.2	ASP 86, ILE 10, LEU 83, HIS 84	LYS 89, LYS 88, GLY 11, GLU 12, GLN 131, VAL 18, LYS 33, GLY 13, ASP 145, THR 14, ALA 31, PHE 80, GLU 81, GLN 85, LEU 134, ASN 132, VAL 64, PHE 82, LYS 20, LEU 298
	PS	−7.7	-	PRO 254, THR 198, LYS 250, VAL 251, ALA 201, LEU 202, ARG 214, PHE 203, THR 218, ARG 200
	STR	−6.5	ARG 214	ARG 200, THR 198, VAL 252, ALA 194, VAL 251, LEU 202, THR 218, PHE 203
P53	LC	−10.5	GLN 260, THR 230, ARG 90, GLN 23, ASN 17, ALA 200, GLU 89, TYR 92, ARG 10	ASN 264, LEU 263, VAL 229, SER 228, CYS 114, LEU 108, PRO 115, GLY 199, HIS 198, ILE 22, ARG 104, LYS 20, ASN 232, GLN 98, THR 94, LYS 97
	PS	−9.2	ILE 21, ARG 90, ASN 232	CYS 114, LEU 103, ALA 200, TYR 92 ARG 203, GLU 13, LYS 18, GLN 38, PRO 231, ARG 10, ASN 17, LYS 20, ILE 22, LEU 100, PHE 16, PRO 115
	STR	−8.9	ASN 17, PHE 16, ILE 21	PRO 115, ARG 104, ARG 203, ARG 10, LYS 20, GLN 23, GLU 89, TYR 92, LEU 100, SER 228

## Data Availability

Data available upon request from the corresponding author.

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
