# Peer review of "Transcriptome Profiling of Cardiac Glycoside Treatment Reveals EGR1 and Downstream Proteins of MAPK/ERK Signaling Pathway in Human Breast Cancer Cells"

_ijms, 2023, doi:10.3390/ijms242115922_

Round 1

Reviewer 1 Report (Previous Reviewer 1)

Comments and Suggestions for Authors

Thanks for revising the manuscript.

Author Response

We sincerely value the feedback and comments provided to enhance the quality of the manuscript. Your input is invaluable, and we are committed to incorporating the necessary improvements.

Reviewer 2 Report (New Reviewer)

Comments and Suggestions for Authors

In this manuscript the authors revealed transcriptional alteration induced by Cardiac Glycoside (GC) including lanatoside C (LC), peruvoside (PS), and strophanthidin (STR) in MCF7 breast cancer cell line. Multiple pathways and genes were involved in this alteration and authors further validated some most significantly expressed genes through RT-PCR, immunoblotting and immunofluorescent imaging. Authors proposed a regulating network involve EGR1 through MAPK and P53-dependent signaling cascade, which mediates cell toxicity upon GC treatment. The manuscript was well written and the results were well supportive to the conclusion. I would recommend publish this manuscript in IJMS if authors address some of the concerns.

Concerns:

(1) Authors tried to establish the inhibitory MAPK to EGR1 signaling pathway in the GC treatment in MCF7 cells, it’s necessary to validate EGR1 is the downstream effort gene in this whole process, if authors could use MAPK inhibitors to further confirm this signaling pathway.

(2) In Figure 2, the figure is too stuffed to see any text. Authors may only selectively label some targeted genes.

(3) In Figure 5, all the * symbols make confusion, each two-group comparison should be labeled clearly. Also Figure 5A is lack of control group legend sign.

(4) In Figure 6, the blots were run poorly. Authors must show original uncut blots.

(5) In Figure 7, the fluorescent images are too small to see, authors have to enlarge the whole panel.

(6) The Font format and sizes of each figure are different, please make them uniformly same.

(7) The references format in the text looked like a combination of two different styles.

(8) The whole manuscript contains several highlights and signs of modification tracks, please remove all of them.

(9) The resolution of all the figures are low, some of texts are difficulty to see.

Author Response

Concerns:

  1. Authors tried to establish the inhibitory MAPK to EGR1 signaling pathway in the GC treatment in MCF7 cells, it’s necessary to validate EGR1 is the downstream effort gene in this whole process, if authors could use MAPK inhibitors to further confirm this signaling pathway.

Response: Thank you for your insightful comment. We acknowledge the importance of confirming EGR1 as the downstream effector gene in the MAPK signaling pathway. However, in the study, our prior focus was to delineate the therapeutic role of CGs on MCF-7 cells and further explore the mechanism of cell death. In the course of our experiments, we identified several genes associated with the MAPK signaling cascade that appear to play a significant role in the pathway leading to cell death. Notably, we observed a substantial downregulation of the EGR1 gene, suggesting it may serve as a key effector in these signaling pathways.

Our hypothesis and conclusion are strengthened by recent studies and reports that support the idea of EGR1 transcription factor regulation through the MAPK signaling mechanism (supporting references cited in MS). This additional context strengthens the rationale behind our focus on EGR1 in the context of CG treatment in MCF-7 cells. We will certainly explore the use of MAPK inhibitors in our upcoming experiments to further validate this signaling pathway.

2. In Figure 2, the figure is too stuffed to see any text. Authors may only selectively label some targeted genes.

Response: We made deliberate revisions to the figure as per your guidance.

3. In Figure 5, all the * symbols make confusion, each two-group comparison should be labeled clearly. Also Figure 5A is lack of control group legend sign.

Response: Thank you for your valuable feedback regarding Figure 5. We have revised the figure as per the suggestion and the statistical significance (* marks) details are provided in the legend.

4. In Figure 6, the blots were run poorly. Authors must show original uncut blots.

Response: We provided the original images for your kind consideration.

5. In Figure 7, the fluorescent images are too small to see, authors have to enlarge the whole panel.

Response: We have revised the figure, meticulously aligning it with your guidance and feedback.

6. The Font format and sizes of each figure are different, please make them uniformly same.

Response: We conscientiously revised the figure to align with your specific guidance and recommendations. However, we encountered challenges with certain graphs and networks created on various platforms, which presented difficulties in achieving seamless integration. We have tried our best to resolve these issues and ensure a harmonious presentation. Note: The modified and updated quality figures are uploaded along with tables in a separate zip folder for more visibility).

7. The references format in the text looked like a combination of two different styles.

Response: Thank you for bringing this to our attention. We have rectified the issue with the references, ensuring a consistent and unified reference style throughout the document.

8. The whole manuscript contains several highlights and signs of modification tracks. Please remove all of them.

Response: We have addressed the issue and removed all highlights and modification tracks from the entire manuscript.

9. The resolution of all the figures are low, some of texts are difficulty to see.

Response: We understand your concern about the resolution of the figures in the manuscript. To address this issue, we have provided the original high-resolution images for your reference, which should enhance the clarity of the figures and make the text more legible.

This manuscript is a resubmission of an earlier submission. The following is a list of the peer review reports and author responses from that submission.

Round 1

Reviewer 1 Report

Comments and Suggestions for Authors

The manuscript by Pavithran et al., presents data uncovering the downstream molecular networks altered with treatment of cardiac glycosides such as Lanatoside C, Peruvoside and Strophanthidine by RNA sequencing. The authors used bioinformatic tools such as KEGG and others to perform their molecular network analyses and uncover different networks in breast cancer cell line models. They uncover several critical pathways that are perturbed with CGs and validate them by qPCR. While the work seems very interesting, it is difficult to draw firm conclusions because the data presented here has little biological insights due to lack of sufficient functional data to validate the downstream molecular pathways identified.

1)      It is well known that RNA seq analyses provides information about altered molecular networks. So, it isn’t surprising that the authors have identified several candidate molecular networks that might be contributing to CG action in cancers. The important question is whether the identified molecular networks have functional consequences. The authors fail to address this question and that is a major limitation of this study. The authors should consider experimentally testing some of their predicted downstream molecular networks/targets.

2)      It is unclear why the authors used ‘untreated’ as control instead of ‘Vehicle’ control. Agents used to dissolve compounds can also influence gene expression. What were the compounds dissolved in? I would suggest the authors to include that in the Methods section.

3)      Font sizes in some of the figures were tiny, fuzzy and unable to read (Ex – Fig 1D, Fig 3). I suggest the authors to increase the font sizes in figures.

Author Response

Comments and Suggestions for Authors

The manuscript by Pavithran et al. presents data uncovering the downstream molecular networks altered with treatment of cardiac glycosides such as Lanatoside C, Peruvoside, and Strophanthidine by RNA sequencing. The authors used bioinformatic tools such as KEGG and others to perform their molecular network analyses and uncover different networks in breast cancer cell line models. They uncover several critical pathways that are perturbed with CGs and validate them by qPCR. While the work seems very interesting, it is difficult to draw firm conclusions because the data presented here has little biological insights due to lack of sufficient functional data to validate the downstream molecular pathways identified.

  • It is well known that RNA seq analyses provides information about altered molecular networks. So, it isn’t surprising that the authors have identified several candidate molecular networks that might be contributing to CG action in cancers. The important question is whether the identified molecular networks have functional consequences. The authors fail to address this question and that is a major limitation of this study. The authors should consider experimentally testing some of their predicted downstream molecular networks/targets.

Response: We greatly appreciate your insightful comment regarding our study. Your valuable suggestion to revise the manuscript with functional implications of the identified molecular networks in the context of CG action in cancers is much valued. In the study, we've successfully identified molecular networks associated with the anticancer activity of CG action. Furthermore, we have conducted an in-depth analysis of its functional implications, employing functional enrichment tools like KEGG and Funrich analysis. In order to further validate our results, we have experimentally tested some of their predicted downstream molecular target expression through RT-PCR. However, the translational level of compound effectiveness needed to be provided.

            We have revised our manuscript, including further assays, Western blot analysis, and immunocytochemistry experiments. We detailed the protein expressional changes and studied its localization at the subcellular level using immunoblotting and immunostaining techniques. These techniques will play a pivotal role in validating and characterizing the predicted downstream molecular networks and their potential targets. In the revised manuscript, we provide protein expression validation on treatment with CG.

  • It is unclear why the authors used ‘untreated’ as control instead of ‘Vehicle’ control. Agents used to dissolve compounds can also influence gene expression. What were the compounds dissolved in? I would suggest the authors to include that in the Methods section.

Response: We are supposed to use vehicle control for our studies, validating the ineffectiveness of the vehicle and ensuring accurate and meaningful interpretation of the results. However, in our preliminary studies involving in-vitro screening, we verified and reported the baseline comparison. We avoided the non-specific effects of ≤ 0.01% DMSO, which was used to dilute cardiac glycoside compounds. We have incorporated the details in the methodology section in keeping with your direction.

  • Font sizes in some of the figures were tiny, fuzzy, and unable to read (e.g., Fig 1D, Fig 3). I suggest the authors to increase the font sizes in figures

Response: We made deliberate revisions to the figure following your guidance.

Reviewer 2 Report

Comments and Suggestions for Authors

The manuscript entitled “Transcriptome profiling of Cardiac Glycoside treatment reveal EGR1 and Downstream proteins of MAPK/ERK Signaling Pathway in Human Breast Cancer Cells” provides new insights. However, the manuscript needs major revision to make the suitable for publication.

1.      Did authors have confirmed the similar research observation in different breast cancer cell lines to avoid cell line specific effects.  

2.      . Authors should also validate RNA-Seq data through Western blotting and Immunoprecipitation (IP).

Author Response

Reviewer#2

Comments and Suggestions for Authors

The manuscript entitled “Transcriptome profiling of Cardiac Glycoside treatment reveal EGR1 and Downstream proteins of MAPK/ERK Signaling Pathway in Human Breast Cancer Cells” provides new insights. However, the manuscript needs major revision to make the suitable for publication.

1). Did authors have confirmed the similar research observation in different breast cancer cell lines to avoid cell line specific effects?

Response:  We appreciate your valuable input in our study. We have carried out preliminary research in different breast cancer cell lines and confirmed the degree of effectiveness specific to the selected MCF-7 breast cancer cell line. In the initial stages of our study, we performed in-vitro screening using a cell culture model, where we systematically evaluated the impact of the compounds as mentioned above across a spectrum of cancer cell lines, including MCF-7 and MDA-MB 231, as well as non-cancerous cell lines.

            Following a rigorous baseline assessment in this study, we specifically focused on the effect of cardiac glycosides on MCF-7 cells, understanding its functional aspects by integrating genomic and molecular studies and techniques.

2). Authors should also validate RNA-Seq data through Western blotting and Immunoprecipitation (IP).

Response:  We're thankful for your thought-provoking comments about our study. In our manuscript revision, we expanded our research by incorporating additional assays, specifically Western blot analysis and immunocytochemistry experiments. Through these different methods, we have provided comprehensive insights into protein expression changes and delved into the subcellular localization of these proteins using immunoblotting and immunostaining techniques. Incorporating these techniques holds significant importance as they serve as crucial components in our endeavor to validate and characterize the predicted downstream molecular networks and identify their potential targets. Within the revised manuscript, we have emphasized providing robust validation of protein expression alterations following treatment with CG.

Round 2

Reviewer 1 Report

Comments and Suggestions for Authors

I appreciate the authors adding more details to the methods section in response to point #2. However, the authors have not satisfactorily addressed my key concern (point #1) and the font size in Fig 1D, 3 are still tiny, fuzzy and unable to read.

Overall, these factors dampen the enthusiasm for this manuscript to be considered for this journal.

Reviewer 2 Report

Comments and Suggestions for Authors

MS looks better after revision. I would recommend for publication in current form.